# Game-Theory-Based Task Offloading and Resource Scheduling in Cloud-Edge Collaborative Systems

**Suzhen Wang \*, Zhongbo Hu, Yongchen Deng and Lisha Hu**

School of Information Technology, Hebei University of Economics and Business, Shijiazhuang 050062, China; huzb0916@163.com (Z.H.); dengyongchen2021@163.com (Y.D.); hulisha@heuet.edu.cn (L.H.)
\* Correspondence: wsuz@163.com

**Abstract:** Task offloading and resource allocation are the major elements of edge computing. A reasonable task offloading strategy and resource allocation scheme can reduce task processing time and save system energy consumption. Most of the current studies on the task migration of edge computing only consider the resource allocation between terminals and edge servers, ignoring the huge computing resources in the cloud center. In order to sufficiently utilize the cloud and edge server resources, we propose a coarse-grained task offloading strategy and intelligent resource matching scheme under Cloud-Edge collaboration. We consider the heterogeneity of mobile devices and inter-channel interference, and we establish the task offloading decision of multiple end-users as a game-theory-based task migration model with the objective of maximizing system utility. In addition, we propose an improved game-theory-based particle swarm optimization algorithm to obtain task offloading strategies. Experimental results show that the proposed scheme outperforms other schemes with respect to latency and energy consumption, and it scales well with increases in the number of mobile devices.

**Keywords:** edge computing; collaborative computation offloading; computation resource allocation; game theory

## 1. Introduction

With the development of IoT and 5G technology, the availability of mobile devices is not limited to wired connections, and a great number of new mobile applications with immersive experiences step into the market, such as virtual reality, connected cars, location awareness, smart cities, etc. Most of these new applications are deployed on mobile devices and are sensitive to real-time communication and intensive computation. Considering the limited resources of mobile devices, it is not feasible to store and process large amounts of multimodal sensory data on these devices. Traditional cloud computing networks [1] need to upload all computing tasks to the cloud center for processing, and it is difficult to avoid problems such as upload delay, network bandwidth, energy loss and data security during transmission. To account for the lack of cloud computing processing, edge computing technology has been proposed and has received a lot of attention. Edge computing [2] is a new computing model that performs computations at the edge of the network close to the user's end device, where the edge is any computing and network resource between the data-generating end and the cloud-centric path. Mobile edge computing (MEC) is the main branch of edge computing. At present, there is no uniform definition of mobile edge computing in academia. The concept given by the European Telecommunications Standardization Institute (ETSI) is that, by deploying edge servers close to mobile users at the network edge, they can use the wireless access network to provide the required services and computing functions nearby. The MEC paradigm provides low-latency, mobility and location-aware support for latency-sensitive applications.

In practice, the processing power and computational resources of edge nodes are limited, and the computational resources allocated to mobile users by edge nodes depend

on the load level of edge nodes (i.e., the number of parallel tasks offloaded to edge nodes). When a great number of mobile devices offload their task demands to the same edge node, it leads to a higher load on the edge node and increased offload task processing latency. A reasonable offloading strategy can improve the edge server's resource utilization and reduce the offloading task processing latency [3] and system energy consumption [4]. Therefore, the offloading strategy of mobile users and the resource scheduling of edge servers are crucial technologies for edge computing. For instance, during the epidemic prevention and control period, with residents' itinerary information and body temperature data monitoring, a reasonable task offloading strategy can allocate data to the best processing location; therefore, the epidemic prevention department can obtain and process data in a timely manner, reducing data processing delays. According to the optimization objectives of offloading decisions, they can be classified into three categories [5]: reducing delay, reducing energy consumption and balancing delay and energy consumption. Liu et al. [6] researched systems that allow parallel computing to be performed on mobile devices and MEC servers. The average delay and energy consumption of task offloading were analyzed using Markov chain theory. The delay optimization problem was mathematically modeled, and a one-dimensional search algorithm was proposed to obtain the optimal offloading strategy. Alam et al. [7] studied the applications of artificial intelligence techniques in task offloading, proposed an autonomous management framework based on Q-learning techniques and solved the problem by Markov decision process modeling and deep reinforcement learning algorithms. Simulation results show that the proposed autonomous deep learning approach can significantly reduce energy consumption. To minimize the delay while saving energy consumption, Li et al. [8] considered the service cache and D2D communication models, and they introduced opportunity networks in a multi-access network; integrated delay and energy consumption as the overall computational overhead; and designed a sequence-game-based suboptimal algorithm to solve the problem.

Most of the existing research on MEC computation offloading only considers task offloading and resource scheduling between mobile devices [9] or between mobile devices and MEC servers [10] without considering the computational resources and processing capabilities of remote cloud servers, and the Cloud-Edge collaboration capability is neglected. With cloud providers aggressively expanding their data centers, cloud resources can be leveraged by deploying high-speed fiber networks. Zhan et al. [11] designed a decentralized computational offloading algorithm to solve the policy optimization problem in the case where users do not disclose their personal information. The problem is described as an observable Markovian decision process, which is solved by a policy-gradient-based deep reinforcement learning (DRL) approach. Tang et al. [12] considered indistinguishable and delay-sensitive tasks and edge load dynamics to formulate the task offloading problem and proposed a distributed algorithm based on model-free deep reinforcement learning. However, none of them considered the computational resources and capacity of cloud servers.

Based on the above problems, this paper studies the Cloud-Edge collaborative system, which is motivated by minimizing the delay, energy consumption and computing cost, and it constitutes the task offloading and resource allocation problem for multiple end users to maximize system utility. The contributions of this paper are summarized as follows:

(1) To address the utility maximization problem, this paper proposes a joint resource allocation and task offloading scheme based on game theory for Cloud-Edge collaboration, including computational resource allocation and task offloading strategy optimization.

(2) The joint task offloading and resource allocation problem is described as mixed-integer nonlinear programming that combines task offloading decisions and resource allocation for offloading users to maximize system utility.

(3) For the joint task offloading and resource allocation problem, an improved particle swarm optimization algorithm based on game theory is proposed to obtain the task offloading strategy, which achieves the Nash equilibrium of the multi-user computational offloading game.

(4)    Other resource allocations and computational offloading schemes are used as comparison schemes for the GTPSO algorithm, and simulation experiments are conducted under different parameters. The results show that the proposed offloading scheme in this paper significantly improves the offloading utility of users.

The rest of this paper is organized as follows: Section 2 reviews related work, Section 3 describes the problem, Section 4 presents the computational offloading resource allocation scheme, Section 5 proposes a multi-user computational offloading game and designs a computational offloading algorithm, Section 6 evaluates the performance of the algorithm and Section 7 concludes the work of this paper.

## 2. Related Work

Many current works have studied the single-user computational offloading problem [13] or the multi-user single-edge server computational offloading problem. Huang et al. [13] proposed a dynamic offloading algorithm based on Lyapunov optimization to improve the performance of mobile edge cloud computing while meeting the execution time of user tasks. Dinh et al. [14] proposed an optimization framework for offloading tasks from a mobile device to multiple MEC servers to minimize total task execution latency and mobile device energy consumption by jointly computing the CPU frequency of the offload and the mobile device. Cao et al. [15] proposed a method for joint computation and communication collaboration in a classical three-node MEC system. A protocol to implement joint computation and communication collaboration was proposed for user delay time-constrained computation on finite length blocks. You et al. [16] considered the resource allocation problem of offloading multiple end devices to a single edge server in MEC and transformed the problem into a convex optimization problem that minimizes the device energy consumption subject to a time delay constraint. Chen et al. [17] studied the computational offloading of MEC in ultra-dense networks by exploiting the software-defined network concept. Guo et al. [18] proposed a distribution strategy to optimize system performance by jointly minimizing the latency time of mobile computing tasks and the operational power consumption of edge cloud servers. Some other works address the multi-user multi-edge server problem. Qiu et al. [19] proposed a new DRL-based online computational offloading scheme, where both blockchain data mining tasks and data processing tasks were considered. Wang et al. [20] studied the resource collaboration of multi-user multi-edge servers under Cloud-Edge collaboration, and the paper proposed an improved artificial bee colony algorithm (ECBL) to match the best edge server for offloading tasks. However, it ignores the processing of task local computation, which causes additional latency and consumption.

Game theory considers all participants as rational users and is used to design decentralized mechanisms that can effectively solve the problem of multiple rational participants making decisions on goals. Game theory is currently applied in both scientific research and life scenarios, providing an effective theoretical basis and solution model for strategic and economic problems. Chen et al. [21] proposed a framework based on Stackelberg's game to maximize the utility of users for the multi-category resource allocation problem of edge servers. Ma et al. [22] designed a service-oriented resource allocation scheme and proposed a three-way round-robin game involving users, edge nodes and service providers. Gu et al. [23] proposed a matching-game-based student project allocation game approach for the joint wireless and computational resource allocation problem. However, these works only consider the resource allocation problem between users and edge servers and do not consider the computational offloading policy problem. Chen et al. [24] considered a multi-user computational offloading problem in a single-channel wireless environment. They described the problem as a decentralized computational offloading game and designed a decentralized computational offloading mechanism. The literature [25] considers a multi-channel environment based on [24] and designs a distributed computational offloading algorithm that achieves Nash equilibrium, and it derives an upper bound on time convergence. Long et al. [26] proposed a multi-objective-computing resource allocation



and computing offloading scheme based on a game-theoretic framework; however, factors such as energy consumption and wireless interference to users were not considered. Hu et al. [27] proposed a game-theoretic-based computational offloading algorithm including a task offloading policy and the transmission of the power control of user devices. The goal of this algorithm is mainly to maximize the number of devices offloaded to the MEC server, and no actual computational resource allocation is performed.

To address the above shortcomings, this paper considers the heterogeneity of mobile devices and inter-channel interference in complex networks with multiple users and multiple edge servers and constitutes a joint task offloading and resource scheduling problem that maximizes system utility, intending to minimize system delay and energy consumption.

## 3. Problem Description

### 3.1. System Model

The Cloud-Edge collaboration system is shown in Figure 1, which contains one cloud computing server, $M$ multi-threaded edge servers and a set of edge servers, which is $M = \{1, 2, \ldots m \ldots M\}$, and each edge server is equipped with a base station that can handle the computing tasks of multiple end-users simultaneously. $N$ end users, the set of users, is denoted as $N = \{1, 2, \ldots n \ldots N\}$, and each user has a non-detachable computationally intensive task $\tau_n$ waiting to be processed. The characteristics of $\tau_n$ are represented as the parameter tuple $\tau_n \triangleq (D_n, C_n, T_n)$, $D_n$ is the input data required by the task, $C_n$ is the total computation required by the task (i.e., the number of CPU cycles) and $T_n$ is the maximum tolerable latency of the task. The computing tasks of mobile users can be processed locally or offloaded to the edge servers for processing. For some special computing tasks that require larger resource consumption and longer task completion cycles, such as data backup for large enterprises, the tasks can be directly migrated to cloud servers for processing and are not considered in the model of this paper. The set of offloading policies is denoted as $S = \left\{ s_n \middle| s_n \in \left\{ s_n^l, s_n^m \right\}, n \in N, m \in M \right\}$; $s_n^l, s_n^m$ denote local computation and edge server computation, respectively; and $s_n^l = 1$ if local execution is selected, and $s_n^m = 1$ if edge server execution is selected.

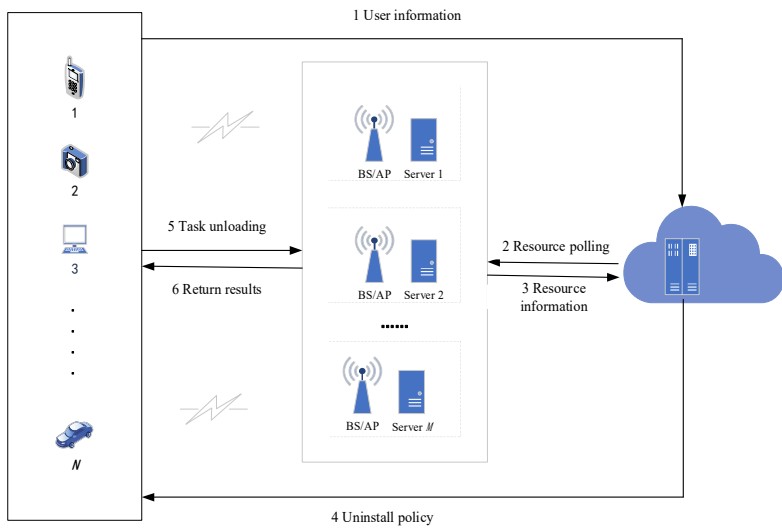

**Figure 1.** Cloud-Edge collaboration system model.

The cloud is responsible for the scheduling of task offloading, making offloading decisions and matching the best edge server for task offloading. The mobile user uploads the data set $I_n$, carrying the computational task information of user $n$ to the cloud center, which collects the entire data set of computational tasks, polls the edge server for resources, retrieves them algorithmically and feeds the obtained resource allocation and offloading policy to the edge server and the end-user. To facilitate experiments and obtain useful data,

we consider quasi-static scenes or simple moving scenes [28,29]. In mobile scenarios, the unloading task needs to be completed before the device is moved to another location. We set the computing power of terminal devices and edge servers to be stable during task processing, and it may change in different time periods.

### 3.2. Computational Models

#### 3.2.1. Local Computing

If the user n processes the computational task locally, the cost overhead consists of only two components: local processing latency and local energy consumption, since no offloading of the task is involved. The local computation time is

$$t_n^l = \frac{C_n}{f_n^l} \tag{1}$$

Local processing energy consumption is

$$e_n^l = C_n \theta_n^l \tag{2}$$

where $f_n^l$ denotes the CPU computing power of the mobile user $n$ (i.e., the number of cycles per second) and $\theta_n^l$ is the energy consumption factor per CPU cycle of the mobile device. The overhead cost of the local processing of computational tasks is

$$E_n^l = \beta_n^t t_n^l + \beta_n^e e_n^l \tag{3}$$

where $\beta_n^t$, $\beta_n^e$ are the weighting factors of delay and energy consumption, satisfying $0 \leq \beta_n^t$, $\beta_n^e \leq 1$ and $\beta_n^t + \beta_n^e = 1$. Different application scenarios have different reliance on delay and energy, and the weighting factors are different.

#### 3.2.2. Edge Computing

In this paper, with the Cloud-Edge-end network system, mobile devices communicate with MEC servers through wireless access point connections, and the MEC and cloud layer communicate by deploying high-speed cables. The non-orthogonal frequency division multiple access techniques are used as a multiple access scheme in the uplink [30]. When user $n$ offloads the computational task $\tau_n$ to the MEC server for processing, the latency includes the user information set to transfer to the cloud center, the processing latency of the cloud center, the uplink transfer latency of the user task and the execution latency of the MEC server. Since the cloud center has sufficient computing resources and the size of the output results is generally much smaller than the input, we ignore the data return latency of the cloud center and edge servers. The set of users offloaded to the MEC server is noted as $N_{off}$.

Set the transmission rate from the end-user to the cloud center to a constant value $r_c$, the channel bandwidth of BS is $B_n$, $p_n$ and $h_n$ are the uplink transmission power and channel gain of user $n$, respectively, $\sigma^2$ is the noise power and $\omega$ indicates the interference brought by other users to user $n$. $\omega = \sum_{i \in N_{off \setminus \{n\}}} p_i h_i$, then the data uplink transmission rate is

$$r_n = B log_2 (1 + \frac{p_n h_n}{\sum_{i \in N_{off \setminus \{n\}}} p_i h_i + \sigma^2}) \tag{4}$$

The edge server processing model latency is

$$t_n^m = t_n^{c,up} + t_n^{c,sear} + t_n^{m,tran} + t_n^{m,exe} \tag{5}$$

where $t_n^{c,up}$ denotes the time of uploading user datasets to the cloud center, $t_n^{c,up} = \frac{I_n}{r_c}$; $t_n^{c,sear}$ is the cloud center processing time, $t_n^{c,sear} = \frac{C_n}{f^c}$; $t_n^{m,trans}$ is the computational task offload time to the edge server, $t_n^{m,tran} = \frac{D_n}{r_n}$; $t_n^{c,sear}$ is the cloud center processing time; and $t_n^{m,exe}$

is the edge server processing time, $t_n^{m,exe} = \frac{C_n}{f_n^m}$, where $f_n^m$ is the computational resource allocated to user $n$ by the edge server.

The energy consumption of the edge server processing task is divided into two parts: data transmission energy consumption and server computation energy consumption, and the energy consumption of the cloud computing center is ignored in this paper. The transmission energy consumption is $e_n^{m,trans} = p_n \left( \frac{I_n}{r_c} + \frac{D_n}{r_n} \right)$; the MEC server computation energy consumption is $e_n^{m,exe} = C_n \theta_n^m$; and the total energy consumption is

$$e_n^m = e_n^{m,trans} + e_n^{m,exe} \tag{6}$$

The overhead cost of offloading the computational tasks to the edge server for processing is

$$E_n^m = \beta_n^t t_n^m + \beta_n^e e_n^m \tag{7}$$

### 3.3. Optimization Goals

In this paper, two metrics of task processing delay and energy consumption are considered, and the problem of resource allocation and task offloading is combined to maximize system utility as the optimization objective. Comparing task latency with the maximum latency $T_n$, more remaining time means higher utility. Comparing the offloading energy consumption with the energy consumption of local processing, higher relative improvements in energy consumption mean higher utility. The utility function of user $n$ for local processing versus offloading to the MEC server for processing is expressed as

$$u_n^l = \beta_n^t \cdot \frac{T_n - t_n^m}{T_n} \tag{8}$$

$$u_n^m = \beta_n^t \cdot \frac{T_n - t_n^m}{T_n} + \beta_n^e \frac{e_n^l - e_n^m}{e_n^l} \tag{9}$$

The utility function of the whole system is defined as the cumulative sum of all user utility values, denoted as

$$U(\mathcal{F}, S) = \sum_{n \in N} u_n \tag{10}$$

where the computational resource allocation $\mathcal{F} = \left\{ 0 < f_n^m \leq F, n \in N_{off} \right\}$, and the offloading policy $S = \left\{ s_n^l, s_n^m \right\}$.

The joint resource allocation and task offloading problem is described as a system utility maximization problem, denoted as

$$\max_{\mathcal{F}, S} U(\mathcal{F}, S) \tag{11}$$

$$s.t. \ C1: \ 0 < f_n^m \leq F, \forall n \in N_{off}$$
$$C2: \ \sum_{n \in N_{off}} f_n^m \leq F^{total}, \forall n \in N_{off}$$
$$C3: \ s_n^l + s_n^m \leq 1, \ \forall n \in N$$
$$C4: \ s_n \in \left\{ s_n^l, s_n^m \right\}, n \in N$$

where C1 is a constraint on the allocation of computing resources available to user $n$ in the MEC server, and C2 is a constraint on the overall computing resources in the edge layer. C3 and C4 are offload policy constraints, and for each computing task, only one offload policy can be selected at a time.

Problem (11) is a mixed nonlinear programming (MINLP) problem, and it is nonconvex and NP-Hard. Using the Tammer decomposition method, the problem can be decomposed into multiple subproblems with separated objectives and constraints. The constraints C1

and C2 for resource allocation and constraints C3 and C4 for offloading decision $S$ are decoupled from each other, so Equation (11) can be transformed into

$$\max_{S}(U^*(S)) \tag{12}$$

*s.t.* C3,C4

where $U^*(S)$ is the optimizing function for the resource allocation problem

$$U^*(S) = \max_{\mathcal{F}}(U(\mathcal{F},S)) \tag{13}$$

*s.t.* C1,C2

Therefore, Problem (11) can be decomposed into a computational resource allocation problem and a task unloading strategy problem, and the optimal unloading decision and resource allocation results can be obtained by iterations.

## 4. Computing Resource Allocation

We propose a game-theoretic-based resource allocation and task offloading scheme for users who choose to offload and solve their resource allocation problem. According to the given task offloading policy $S$ that satisfies the constraints, the utility of the set of users offloading to MEC, $U_m(\mathcal{F},S)$, is obtained using the expression $u_n^m$ for the utility function, denoted as

$$U_m(\mathcal{F},S) = \min_{\mathcal{F}} \sum_{n \in N_{off}} [\beta_n^t \frac{(T_n - t_n^m)}{T_n} + \beta_n^e \frac{(e_n^l - e_n^m)}{e_n^l}] \tag{14}$$

To maximize the utility function $U_m(\mathcal{F},S)$, which is equivalent to minimizing the unloading overhead, it can be expressed as

$$\min_{\mathcal{F}} \sum_{n \in N_{off}} (\beta_n^t t_n^m + \beta_n^e e_n^m) \tag{15}$$

When a user offloads its computational tasks to the MEC server, the computational resource allocation problem is expressed as

$$\min_{\mathcal{F}} \sum_{n \in N_{off}} \{\beta_n^t (\frac{I_n}{r_c} + \frac{D_n}{r_n} + \frac{C_n}{f^c} + \frac{C_n}{f_n^m}) + \beta_n^e [p_n(\frac{I_n}{r_c} + \frac{D_n}{r_n}) + C_n\theta_n^m]\} \tag{16}$$

$$s.t.\ C1 : 0 < f_n^m \le F, \forall n \in N_{off}$$
$$C2 : \sum_{n \in N_{off}} f_n^m \le F^{total}, \forall n \in N_{off}$$

The objective function in Problem (15) is denoted as $\varphi(\mathcal{F},S)$, and the first and second-order derivatives of $\varphi(\mathcal{F},S)$ with respect to $f_n^m$ are obtained.

$$\frac{\partial \varphi(\mathcal{F},S)}{\partial f_n^m} = -\frac{\beta_n^t C_n}{(f_n^m)^2}\ , \forall n \in N_{off} \tag{17}$$

$$\frac{\partial^2 \varphi(\mathcal{F},S)}{\partial(f_n^m)^2} = \frac{2\beta_n^t C_n}{(f_n^m)^3}\ , \forall n \in N_{off} \tag{18}$$

The second-order derivative of $\varphi(\mathcal{F},S)$ with respect to $f_n^m$ is constantly greater than zero, and the constraints C1 and C2 are both convex. Therefore, Problem (16) is a convex optimization problem satisfying the Slater condition, and the KKT condition can be used to find the optimal computational resource allocation $\mathcal{F}^*$. Formulating Problem (16) as a Lagrange function results in:

$$L(f_n^m, \lambda, \eta) = \sum_{n \in N_{off}} \{ \beta_n^t (\frac{I_n}{r_c} + \frac{D_n}{r_n} + \frac{C_n}{f^c} + \frac{C_n}{f_n^m}) + \beta_n^e [p_n(\frac{I_n}{r_c} + \frac{D_n}{r_n}) + C_n \theta_n^m]\} + \lambda(f_n^m - F) + \eta(\sum_{n \in N_{off}} f_n^m - MF) \quad (19)$$

where $\lambda$ and $\eta$ are Lagrange multipliers associated with computational resource constraints satisfying $\lambda \geq 0$ and $\eta \geq 0$. Solving for the first-order derivative of $L(f_n^m, \lambda, \eta)$ with respect to $f_n^m$ yields

$$f_n^{m*} = \sqrt{\frac{\beta_n^t C_n}{\lambda^* + \eta^*}}, \ \forall n \in N_{off} \quad (20)$$

Since $f_n^{m*}$ is coupled with Lagrange multipliers $\lambda^*$ and $\eta^*$, the gradient descent method is used to iteratively update $\lambda, \eta$ until the computational resource constraints C1 and C2 are satisfied, and the optimal computational resource allocation solution $f_n^{m*}$ is obtained. The process is shown in Algorithm 1.

---

**Algorithm 1:** MEC computing resource allocation scheme

---

1: **Initialization:** very small tolerance $\varepsilon > 0$, $\lambda = \lambda^{max}$, $\eta = \eta^{max}$
2: **While** $f_n^{m*} - f_n^m > \varepsilon$ do
3:      set $f_n^m = f_n^{m*}$
4:      compute $f_n^m$ according to substitute $\lambda, \eta$ into (20).
5:      **If** $f_n^m < F$, update $\lambda = \lambda - \Delta\lambda$
          $\sum_{n \in N_{off}} f_n^m < MF$, update $\eta = \eta - \Delta\eta$
6: **end while**
7: The optimal computation resource allocation scheme can be derived by substituting $\lambda, \eta$ into (20).
8: **Output:** $\mathcal{F} = \left\{ f_n^{m*} \middle| 0 < f_n^{m*} \leq F; \sum_{n \in N_{off}} f_n^m < MF, n \in N_{off} \right\}$

---

## 5. Task Unloading Strategy

In this section, a game theory approach and particle swarm optimization algorithm are used to solve the task unloading policy problem, and an improved game-theory-based particle swarm optimization (GTPSO) algorithm is proposed to obtain the optimal task unloading policy.

### 5.1. Multi-User Task Offloading Game

The task unloading strategy is defined as a game-theoretic problem, denoted as $G = \left\{ N, (S_n)_{n \in N}, (u_n)_{n \in N} \right\}$, where the mobile user $N$ denotes the participant of the game; $S$ is the set of task unloading strategies of the user; each user has $(M + 1)$ choices of task processing strategies; $s_n \in \left\{ s_n^l, s_n^1, \cdots s_n^M \right\}$. $u_{(S_n, S_{-n})}$ is the utility function of user $n$; and $s_{-n}$ denotes the set of offloading policies for users other than user $n$, $s_{-n} = (s_1, \cdots, s_{n-1}, s_{n+1}, \cdots s_N)$. Supposing $N$ rational mobile users, each of whom chooses a task processing strategy that optimizes its own utility function, the resulting function is

$$\max_{s_n^*}(u(s_n, s_{-n})) = s_n^l u_n^l + s_n^m u_n^m \quad (21)$$

Nash equilibrium (NE) is an important concept in game theory, which refers to the situation where any one participant in a non-cooperative game system chooses the combination of optimal strategies given that the strategies of other participants are determined.

**Definition 1.** *If given the set* $S^* = \left\{ s_1^*, \cdots s_n^*, \cdots s_N^* \right\}$ *of unloading strategies, for* $\forall n \in N$, *all have*

$$u(s_n^*, s_{-n}^*) \geq u(s_n, s_{-n}^*), s_n^* \in \left\{ s_n^l, s_n^m \right\} \quad (22)$$

Then, the set of strategies $S^*$ is the NE of the game $G$. A Nash equilibrium is a state that makes the system stable, and the users in the set of strategies have no incentive to leave the NE.

**Definition 2.** *If there exists a potential function $\psi(s)$, when the unloading strategy of the participating users in the game $G$ is changed unilaterally from $s_n$ to $s'_n$, we can obtain*

$$u(s_n, s^*_{-n}) - u(s'_n, s^*_{-n}) = \psi(s_n, s_{-n}) - \psi(s'_n, s_{-n}), \forall n \in N \tag{23}$$

Then, the game $G$ is a potential game. According to the properties of potential games, there exists a Nash equilibrium for a potential game with a finite set of strategies and finite improvement, which can be reached in a finite number of improvements.

**Theorem 1** . *The game $G$ is a potential game with the potential function shown in Equation (24) and can reach a Nash equilibrium within a finite number of improvements.*

$$\psi(s) = s_n^l \sum_{i=1}^{N} u_i^l + \left(1 - s_n^l\right)\left(s_n^m u_n^m + \sum_{i=1 \&\& i \neq n}^{N} u_i^l\right) \tag{24}$$

**Proof.** It satisfies $s_n = s_n^l = 1; s_n = s_n^m = 0$ when the user performs computational tasks locally. $s_n = s_n^l = 0; s_n = s_n^m = 1$ when the user processes computational tasks on the edge server. For user $n$, $(\forall n \in \text{N})$, the potential function should satisfy Equation (24) when the user's offloading decision is updated from $s_n$ to $s'_n$. We consider the following three cases:

(1) The offload policy for mobile user $n$ is updated from local processing $s_n^l$ to edge server processing $s_n^m$. We can obtain

$$\begin{aligned}\psi\left(s_n^l, s^*_{-n}\right) - \psi(s_n^m, s^*_{-n}) \ &= \sum_{i=1}^{N} u_i^l - u_n^m - \sum_{i=1 \&\& i \neq n}^{N} u_i^l \\ &= u_n^l - u_n^m \\ &= u\left(s_n^l, s^*_{-n}\right) - u(s_n^m, s^*_{-n})\end{aligned}$$

(2) The offload policy for mobile user n is updated from offload to edge server processing $s_n^m$ to local processing $s_n^l$. We can obtain

$$\begin{aligned}\psi(s_n^m, s^*_{-n}) - \psi\left(s_n^l, s^*_{-n}\right) \ &= u_n^m + \sum_{i=1 \&\& i \neq n}^{N} u_i^l - \sum_{i=1}^{N} u_i^l \\ &= u_n^m - u_n^l \\ &= u(s_n^m, s^*_{-n}) - u\left(s_n^l, s^*_{-n}\right)\end{aligned}$$

(3) The offload policy for mobile user n is updated by edge server $s_n^m$ to edge server $s_n^{m'}$ for processing. We can obtain

$$\begin{aligned}\psi(s_n^m, s^*_{-n}) - \psi\left(s_n^{m'}, s^*_{-n}\right) \ &= \left(s_n^m u_n^m + \sum_{i=1 \&\& i \neq n}^{N} u_i^l\right) - \left(s_n^{m'} u_n^{m'} + \sum_{i=1 \&\& i \neq n}^{N} u_i^l\right) \\ &= u_n^m - u_n^{m'} \\ &= u(s_n^m, s^*_{-n}) - u\left(s_n^{m'}, s^*_{-n}\right)\end{aligned}$$

Combining the above cases, we can obtain that the change in the potential function always satisfies Equation (23) for changes in the user $n$ offloading decision. Therefore, the game $G$ is a potential game that can reach Nash equilibrium within a finite number of improvements, further proving that there exists a set of offloading policies making the utility function of $N$ rational mobile users optimal in this paper's complex network of multi-user multi-edge servers under the Cloud-Edge collaborative architecture. □

*5.2. GTPSO Algorithm*

The particle swarm optimization (PSO) algorithm is a stochastic search algorithm for finding optimal solutions among multiple participating individuals through information sharing and mutual collaboration [31]. In this section, an improved game-theory-based particle swarm optimization algorithm (GTPSO) is designed for the task offloading policy problem in Section 5.1. The improved particle swarm optimization algorithm is first used to obtain the preprocessing unloading policy, and then the computational resource allocation is optimized. The two processes of the unloading strategy and computational resources are iteratively updated with each other until convergence to maximize system utility.

5.2.1. Pre-Processing Offload Strategy

1.   Particle encoding

Combined with the system in this paper, an integer encoding mechanism is used, and the particle swarm size is set to $K$. Each particle represents a collection of offloading policies. The particle element represents the location of the edge server to be offloaded by the current computing task, and the number of particle elements is $N$. The particle elements take the values of integers from $1 \sim M$. The velocity vector of particle $k = \{task_1, \cdots task_n, \cdots task_N\}$ is $V_k = \{v_{k1}, v_{k2}, \cdots, v_{kN}\}$; the position vector is $X_k = \{x_{k1}, x_{k2}, \cdots, x_{kN}\}$; $P_{best}$ denotes the individual optimal solution of the current particle; and $G_{best}$ denotes the global optimal solution of all particles.

2.   Fitness function

The fitness function of a particle is typically the objective function of the problem being solved, and in this paper, the system overhead of offloading tasks to the edge servers for processing is used as the fitness function, i.e., the total system cost of assigning tasks to different edge servers, as in Equation (25).

$$Fitness(V) = \min_{x_n} \sum_{m=1}^{M} \sum_{n=1}^{N} E_n^m \tag{25}$$

$$s.t. \text{C1}: \ 0 \le E_n^m \le E_{total}^m \le E_{max}$$
$$\text{C2}: \ 0 \le f_n^m \le f_{total}^m \le f_{max}$$

where $x_n$ denotes the server number specifically assigned to the task, C1 denotes the energy constraint offloaded to the task numbered $m$ and C2 denotes the computing resource constraint assigned to the task numbered $m$.

Since we ignore task waiting time and consider the existence of load imbalance caused by multiple users offloading to an edge server, and since the edge server is rational, we can set the penalty function; therefore, the total system cost increases to relieve the load pressure when tasks are offloaded to the edge server with a high load.

$$Fitness(V) = \min_{x_n} \sum_{m=1}^{M} \sum_{n=1}^{N} E_n^m + penalty(V) \tag{26}$$

$$penalty(X) = g * \sum_{m=1}^{M} \sum_{n=1}^{N} (E^m - E_{max})$$

Equation (26) indicates that, if the energy consumption of the task load offloaded to task number $m$ is higher than the maximum energy consumption of the server, a penalty function is set to increase the system cost, where $X$ denotes the offloading vector solved for and $g$ denotes the penalty factor.

3.   Algorithm Process

Input:

(1)   User set $N = \{1, 2, \ldots N\}$, MEC server set $M = \{1, 2, \ldots M\}$, task set $\tau_n = (D_n, C_n, T_n)$.

(2)   Algorithm control parameters: maximum number of iterations *maxGen*, velocity boundary $v \in [-M, M]$, position boundary $x \in [1, M]$, initial inertia factor $w$ and penalty factor $g$.

Initialization:

(1)   The position vector $X_k$ of each particle and the velocity vector $V_k$.
(2)   The fitness function value $Fitness(V)$ is initialized and updated as the iteration progresses.
(3)   Initialize the individual optimal solution and the global optimal solution of the particle, set the current position of the particle as the individual optimal solution $P_{best}$ and set the position of the particle with the smallest fitness value as the global optimal solution $G_{best}$.

Iterative Process:

(1)   Let the number of iterations $y = 1$.
(2)   While $y \leq maxGen$.
(3)   Update the velocity; each dimension in particle k independently goes to update the velocity $V_k$. If $|V_k| > M$. Then, when $V_k > 0$, let $V_k = M$, and when $V_k < 0$, let $V_k = -M$. The updated formula of particle velocity is:

$$V_k = w_{max} * V_k + c_1 * r_1(P_{best} - X_k) + c_2 * r_2(G_{best} - X_k) \tag{27}$$

where $c_1$ and $c_2$ are learning factors and $c_1 = c_2 = 1.5$, and $r_1$ and $r_2$ are random numbers from 0~1.

(4)   Update the position. Particle $k$ updates the position $X_k$ independently based on the velocity information. If the value of $X_k$ is greater than $M$, then let $X_k = M$. The particle updated position equation is:

$$X_k = X_k + V_k \tag{28}$$

(5)   Update inertia weights. The fixed inertia weight values easily lead the algorithm to fall into partial optimality. Consider changing the fixed inertia weights in the standard PSO algorithm to a dynamic adjustment strategy to avoid falling into partial optimality and to obtain a better solution to the problem. To ensure that the algorithm starts with a global search in large steps, a large value is initially assigned to $w$. As the number of iterations increases, w gradually decreases; therefore, the solution of the problem can be traded off between the local optimum and the global optimum. The weights are calculated as in Equation (29):

$$w = w_{max} - w_{max} * \frac{y}{Y} \tag{29}$$

(6)   Update the particle optimal allocation and global optimal allocation. All particles are calculated according to Equation (26) after iteration, and if the updated fitness function value is smaller than the current value, the particle's individual optimal allocation $P_{best}$ and global optimal solution $G_{best}$ are updated.
(7)   Update the number of iterations, $y = y + 1$.
Output: Optimal allocation vector $X_k = G_{best}$ and minimum delay $Fitness(V)$.

After multiple iterations, when the number of iterations reaches the maximum or the optimal solution does not change after the iterations, the best allocation vector $G_{best}$ is obtained, which is the optimal task offloading policy under the current computing resources, and the obtained task offloading policy is used as the preprocessing task offloading policy.

5.2.2. Policy Update Process

The task offloading policy obtained from the improved particle swarm optimization algorithm is used as the preprocessing task offloading policy, and the two processes are iterated over each other until convergence to maximize system utility using Algorithm

1 to optimize resource allocation under the given offloading policy. Each iteration of the algorithm consists of two processes:

(1) Resource allocation optimization: The user uses Algorithm 1 to optimize resource allocation according to the current offload policy and calculates the corresponding utility values for different offload policies.

(2) Policy update competitions: Based on the optimized computational resources, calculate the utility of each user with different uninstallation policies. The users who can improve their utility compete for the policy update opportunity in a distributed form, and the user with the largest utility improvement updates the uninstallation policy, whereas other users keep the original uninstallation policy and wait for the next round of decision updates. Using the finite improvement property of the potential game, only one user with the maximum utility improvement is allowed to update the uninstallation strategy in each iteration. The iteration terminates when the Nash equilibrium is reached and when all users have no incentive to change their uninstallation strategies. The uninstallation policy that maximizes the utility of the system is obtained.

The process is shown in Algorithm 2.

---

**Algorithm 2**: GTPSO

---

1: **Input**: user set $N = \{1, 2, \ldots N\}$, $M = \{1, 2, \ldots M\}$, $\tau_n = (D_n, C_n, T_n)$, $n \in N$;
   $maxGen$, $v \in [-M, M]$, $x \in [1, M]$, $w$, $g$.
2: **For each particle**
3:   Initialize position $X_k$, $V_k$, $Fitness(V)$, $P_{best}$, $G_{best}$
4: **End For**
5: **Iteration** $y = 1$
6: **DO**
7:   Update the $V_k$ by (27) and $X_k$ by (28)
8:   Update the $w$ by (29)
9:   Evaluate particle $k$
10:   **If** $Fit\,(X_k)\ <\ Fit\,(P_{bestk})$
11:    $P_{bestk} = X_k$
12:   **End if**
13:   **If** $Fit(P_{bestk})\ <\ Fit(G_{best})$
14:    $G_{best} = P_{bestk}$
15:   **End if**
16:   $k = k + 1$
17: **WHILE** maximum iterations or optimal solution are not changed
18: **Output**: Pre task offloading strategy $S^*$
19: $t \leftarrow t+1$
20: **while** $S^*(t) \neq S^*(t - 1)$ **do**
21:   $S^* = S^*(t - 1)$, **set** $n = 1$
22:   **while** $n \leq N$ **do**
23:    calculate $u_n^l\left(s_n^l, s_{-n}^*(t - 1)\right)$ by (8)
24:    calculate $f_n^{m*}$ and $u_n^m(s_n^m, s_{-n}^*(t - 1))$ by algorithm 1 and (9), respectively
25:    compute the best response $\Delta_n(t)$
26:    $n = n + 1$
27:   **end while**
28:   **for each user** $n, n \in N$ **do**
29:    **if** user $n$ wins in the $t-th$ iteration,
30:     **then** update $s_n(t)$
31:    **else** $s_n(t) = s_n(t - 1)$
32:   **end for**
33:   $t = t + 1$
34: **end while**
35: **Output**: Optimal computation resource allocation $\mathcal{F}^*$ and offloading strategy $S^*$

---

## 6. Experimental Results and Analysis

### 6.1. Experimental Setup

To verify the performance of the proposed task offloading scheme, we performed experimental simulations on a Windows PC using Matlab2021 data analysis and Python programming software. We considered face recognition or body temperature monitoring as the user's computational task and evaluated the overhead of task offloading for the end-user based on the utility value of the task processing system. Since there is no standardized experimental platform or data for edge computing, we used self-combining data for our experiments. Without a loss of generality, we used a centralized cloud server with an MEC collaboration scheme under a fiber–radio hybrid network. The communication and computational parameters in this paper were set concerning the literature [32,33], and the uplink channel gain was generated by the path loss model $L(dB) = 140.7 + 36.7 \text{loglog}_{10} d_{[km]}$. The standard deviation of the log-normal shading was set to 8 dB. Unless otherwise stated, we chose the input data amount from [200~600] KB randomly, the required CPU cycle was 1 GHz and the time and energy weighting factors were set to $\beta_n^t = 0.2$ and $\beta_n^e = 0.8$, respectively. Other partial parameters were set as shown in Table 1.

**Table 1.** Experimental parameters.

| Experimental Parameter | Numerical Value |
|---|---|
| Cloud center CPU frequency | 20 GHz |
| Edge server CPU frequency | 10 GHz |
| User CPU frequency | 1 GHz |
| Cloud transmission rate $r_c$ | 1 us/bit |
| System bandwidth B | 20 MHz |
| Uplink power $p$ | 20 dBm |
| Channel gain $h$ | 0.6~0.8 |
| Noise power $\sigma^2$ | −100 dBm |
| $\theta_n^l$ | $10^{-9}$ cycle/J |
| $\theta_n^m$ | $10^{-10}$ cycle/J |
| Penalty coefficient $g$ | $10^{-2}$ |
| Inertia weight $w_{max}$ | 0.8 |

The following scenarios were used as comparison scenarios for GTPSO:

1. Exhaustive: This is a brute-force method that finds the optimal offloading scheduling solution via an exhaustive search of over $2^n$ possible decisions. Since the computational complexity of this method is very high, its performance is only evaluated in a small network setting.

2. Task offloading by the particle swarm optimization algorithm (TOPSO): Using the particle swarm optimization algorithm in [34] for task offloading and introducing the cloud center for offloading scheduling, the TOPSO scheme does not consider the resource allocation scheme.

3. Joint Greedy Offloading and Resource Allocation (JGORA): All tasks are offloaded [35], and each offloaded user greedily selects the subchannel with the highest channel gain until all users are admitted or until all subchannels are occupied. The JGORA scheme does not account for the cloud computing processing model.

4. ECBL: The literature [20] proposes an improved artificial bee colony algorithm to find the optimal allocation scheme. The ECBL scheme considers the cloud-side collaborative system but does not consider the task local processing scheme.

Performance Evaluation

To verify the suboptimality of the algorithm, GTPSO was compared with other schemes, and the system utility values of all schemes at different loads for $N = 10$ and $M = 4$ are given in Figure 2. We found that the exhaustive method has the highest utility value, and the GTPSO scheme has significantly better utility values than the other schemes.

In addition, with increases in computational load, the computation required for the task increases, which causes greater latency and energy consumption, and the systematic utility value decreases overall. There was a performance improvement of 15%, 10% and 7.1% compared to the TOPSO, JGORA and ECBL schemes, respectively.

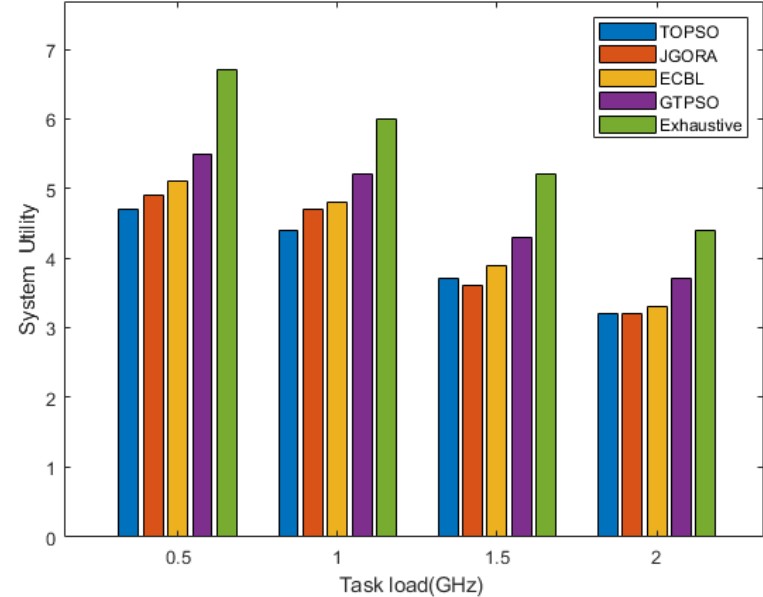

**Figure 2.** Comparison of system utility.

The running time of the program for each scheme is shown in Table 2. The running time of the exhaustive method greatly exceeds that of the other schemes; therefore, although the exhaustive method yields the optimal solution, the running time is unacceptable. The JGORA scheme does not consider the cloud computing processing model and has a higher running time than the other schemes. The running time of the GTPSO scheme, although higher, is maintained at the millisecond level and is negligible compared to the total time (at the second level) of edge computing task execution. Therefore, the GTPSO scheme is the more practical suboptimal solution with respect to combined performance and runtime.

**Table 2.** Runtime of schemes (ms).

| Scheme | Time |
|--------|------|
| TOPSO | $3.7 \pm 0.2$ |
| JGORA | $21.2 \pm 2.5$ |
| ECBL | $5.2 \pm 0.7$ |
| GTPSO | $11.3 \pm 0.4$ |
| Exhaustive | $3627 \pm 36$ |

Figure 3 gives a comparison of the system utilities of different schemes at the termination of an iteration with a different number of users. The system utility values of all schemes increase with increases in the number of users. Among all the schemes, the GTPSO scheme has the best performance because the GTPSO scheme can fully utilize the MEC and cloud computing resources. The cloud center is responsible for the overall algorithm scheduling, and the MEC server is responsible for the endpoint task processing, which jointly optimize the computing resources and offloading strategy.

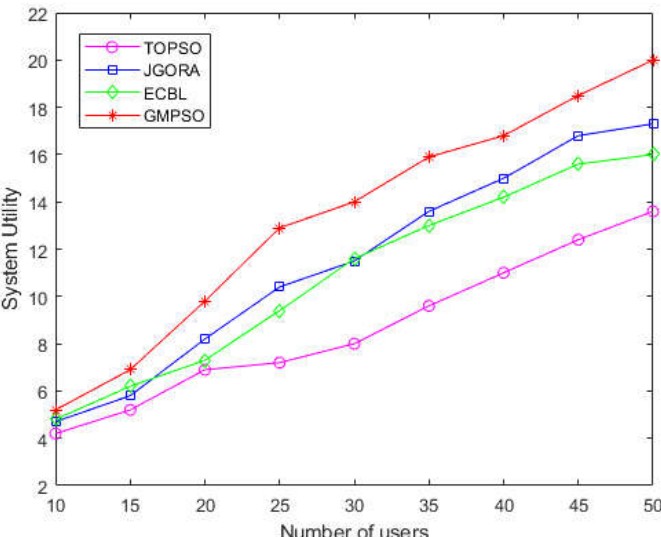

**Figure 3.** System utility against different numbers of users.

Figure 4 gives the average task latency for each scheme at different computational loads for *N* = 20 and *M* = 6. The task processing latency increases as the computational load increases. Among all the schemes, GTPSO has the smallest latency, followed by the TOPSO scheme, because the TOPSO scheme does not consider the optimization of computational resource allocation. The JGORA scheme has the largest latency because it can only offload tasks to resource-constrained edge servers without considering the scheduling capability of cloud resources. As the computational load continues to increase, the GTPSO scheme has more significant advantages.

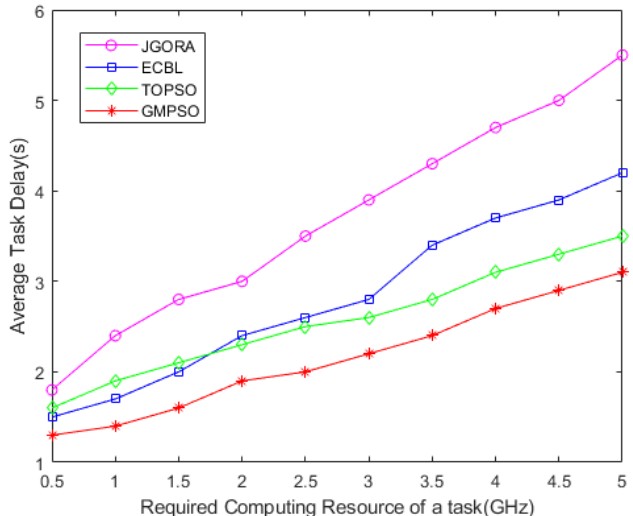

**Figure 4.** Average task delay against computation load.

Figure 5 gives a comparison of the system utility convergence of the GTPSO scheme with the standard particle swarm algorithm scheme (SPSO) and TOPSO scheme in the preprocessing strategy stage to evaluate the impact of the improved particle swarm algorithm on the global system utility value. With increases in the number of iterations, the user's utility value increases. The TOPSO scheme converges firstly due to ignoring resource allocation, but the system utility value is lower. The SPSO scheme does not consider the computational load caused by multiple end tasks offloaded to the same MEC server, and the system utility value is lower than that of the GTPSO scheme. The GTPSO scheme has

about 8% and 15% performance improvement compared to the TPSO and TOPSO schemes.

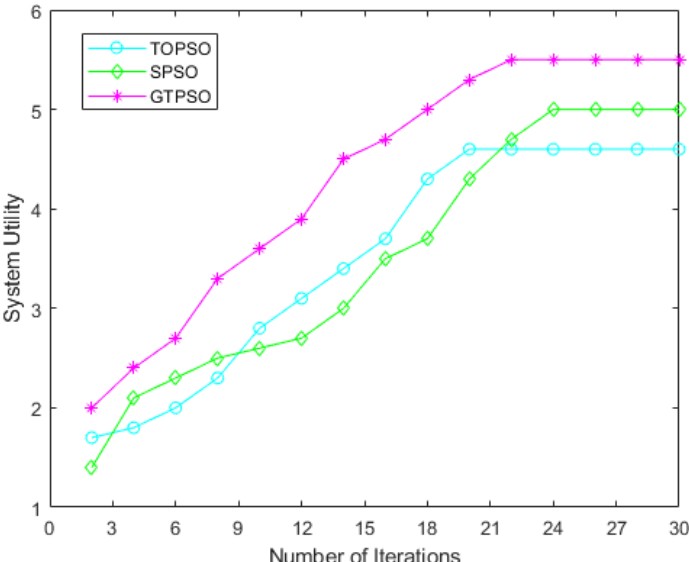

**Figure 5.** Import of system utility by PSO.

Figure 6 gives the system convergence of the GTPSO scheme for $N = 20$ and $M = 6$ with different amounts of computing resources in the cloud center. The system utility value gradually increases with the number of iterations and finally converges to stability. As the amount of cloud computing resources increases, the cloud center performs the policy schedule faster, the system latency decreases and the utility value increases. Each iteration of the GTPSO scheme optimizes the two processes of computing resources and offloading policies to increase system utility until the system reaches Nash equilibrium.

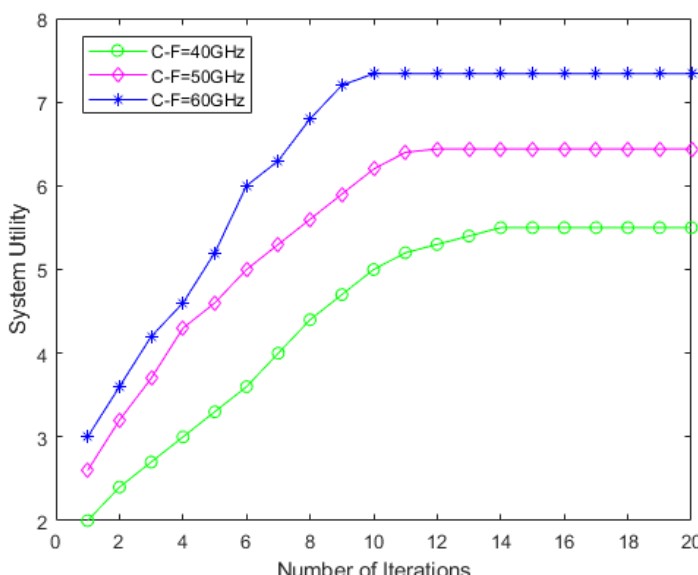

**Figure 6.** Impact of cloud resources.

Figure 7 gives the system convergence of the GTPSO scheme for $N = 20$ and $M = 6$ with different amounts of edge server computing resources. As the amount of MEC server resources increases, the computational resources available for offloading tasks become larger, and the system utility value gradually increases with the number of iterations until it stabilizes. In addition, as the computing resources of the cloud and edge servers become larger, the system utility value becomes higher, the number of iterations becomes fewer

and the convergence becomes faster. In addition, the computing resources of the cloud server have more influence on the global utility value.

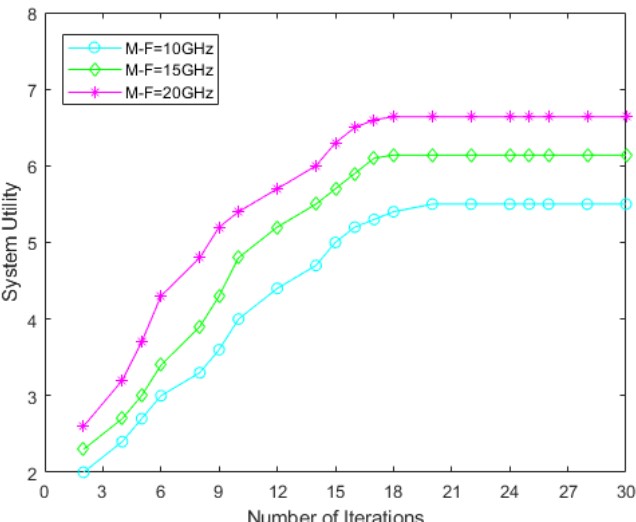

**Figure 7.** Impact of MEC resources.

Figure 8 gives the time delay and energy consumption for each scenario concerning the time delay weight and energy consumption weight for $N = 20$ and $M = 6$. Figure 8a shows the average task latency versus time weight. $\beta_n^t$ varies from 0 to 1, and the latency of all schemes decreases as $\beta_n^t$ increases. When $\beta_n^t$ increases to a certain threshold, the average delay decreases to a certain level and then converges. The delay of the GTPSO scheme is always kept to a minimum. Figure 8b shows the relationship between the average task energy consumption and the energy consumption weighting factor. The energy consumption of each scheme decreases as $\beta_n^e$ increases. The JGORA scheme converges to a level close to 0 at $\beta_n^e$, and its power control method keeps the user in a smaller range, which is more suitable for scenarios with high energy consumption requirements. The GTPSO scheme has the smallest energy consumption at $\beta_n^e \leq 0.4$. The GTPSO scheme makes a trade-off between latency and energy consumption to reduce the energy consumption of user devices while ensuring low latency.

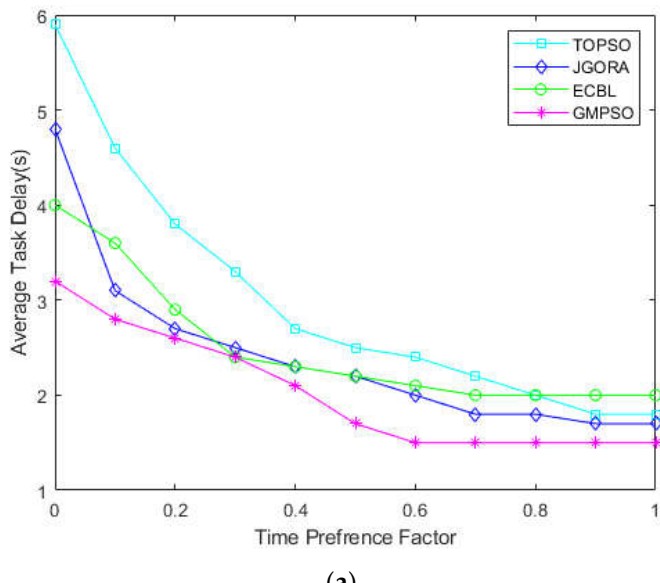

(**a**)

**Figure 8.** *Cont.*

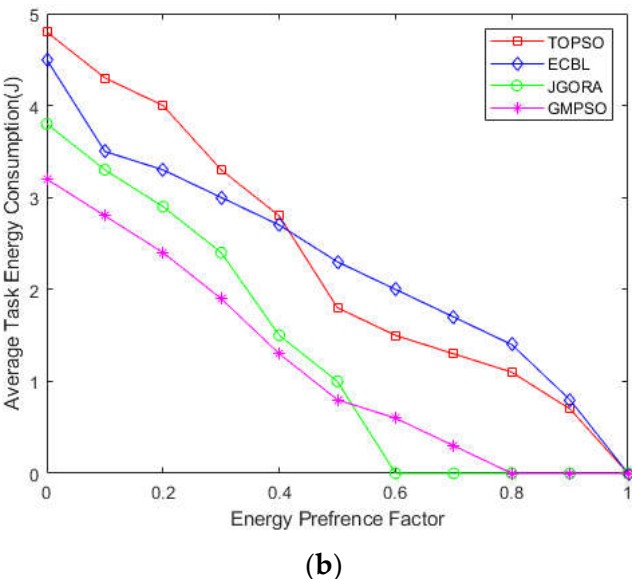

**(b)**

**Figure 8.** Delay and energy consumption versus weighting factors: (**a**) Average task latency at different $\beta_n^t$; (**b**) Average task energy consumption at different $\beta_n^e$.

## 7. Conclusions

In this paper, we study the joint task offloading and resource allocation problem in a multi-user network under a cloud-side collaborative architecture with the objective of optimizing system utility. Since the problem is an MINLP problem, it is difficult to obtain an optimal solution, which is decomposed into two sub-problems of computational resources and task offloading, and the computational resource allocation problem is solved using the KKT condition. For the task offloading policy optimization problem, a GTPSO algorithm is proposed to reduce time complexity while ensuring better performance. The experimental results show that the GTPSO scheme has good performance with respect to delay, energy consumption and system utility.

Since the algorithm proposed in this paper only considers the static scenarios of users, users may dynamically leave the system during computing offloading or resource allocation. This affects the performance of the solution, and it requires all end users to upload information to the cloud center, which causes user information to be safe. In future research, we will study end-user mobility and combine federated learning technology with edge computing technology to protect user privacy information during task processing.

**Author Contributions:** Conceptualization, S.W., Z.H., Y.D. and L.H.; Methodology, S.W., Z.H. and L.H.; Project administration, S.W.; Software, Z.H.; Validation, Z.H.; Visualization, S.W., Z.H. and Y.D.; Writing—original draft, Z.H.; Writing—review & editing, S.W., Z.H., Y.D. and L.H. All authors have read and agreed to the published version of the manuscript.

**Funding:** This research was partially supported by the Natural Science Foundation Project of Hebei Province, China (No. F2021207005).

**Institutional Review Board Statement:** Not applicable.

**Informed Consent Statement:** Not applicable.

**Data Availability Statement:** The data that support the findings of this study are available on request from the corresponding author.

**Conflicts of Interest:** The authors declare no conflict of interest.

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
