# Peer review of "Game-Theory-Based Task Offloading and Resource Scheduling in Cloud-Edge Collaborative Systems"

_applsci, doi:10.3390/app12126154_

Round 1

Reviewer 1 Report

The paper presents a very interesting and innovative approach. It presents a fully detailed background research and extensive literature review. The soundness of the proposed model is really good. The conclusions are in line with the results. 

No recommendations for improvement.

Author Response

Response to Reviewer 1 Comments

      Thank you very much for your review of our paper and for your support of the content in this paper.

Reviewer 2 Report

The paper presents the application of game theory and particle swarm optimization to task offloading and resource scheduling in edge computing systems. The method outperforms existing approaches but the paper should be improved to be more readable and understandable:

1.  Some real-life examples would show the essence of the problem and would outline the motivation to this research.

2. The paper requires a lot of editorial corrections e.g. a lot of equations are not readable and require justification. 

3. More exhausting discussion about the advantages and disadvantages of the presented approach would be interesting. The method is heuristic thus it is not guaranteed that it will give good results in all cases. 

Author Response

Response to Reviewer 2 Comments

 We are very grateful to the reviewers for their comments on the article, and we have made careful revisions to the corresponding comments.

Point 1:

Some real-life examples would show the essence of the problem and would outline the motivation to this research.

Response 1:

In lines 27-31 of the introductory part, we take the data processing during the most urgent period of epidemic prevention and control as an example. “For instance, during the epidemic prevention and control period, residents' itinerary information and body temperature data monitoring, a reasonable task offloading strategy can allocate data to the best processing location, so that the epidemic prevention department can obtain and process data in a timely manner, reducing data processing delays.”

 In the fourth paragraph of the introduction, the motivation for the research content of this paper is presented. “Based on the above problems, this paper studies the cloud-edge collaborative system, which is motivated by minimizing the delay, energy consumption, and computing cost, and constitutes the task offloading and resource allocation problem for multiple end users to maximize the system utility.”

Point 2:

The paper requires a lot of editorial corrections e.g. a lot of equations are not readable and require justification.

 Response 2:

We have carefully checked the formula section of the article and made some changes.

Point 3:

More exhausting discussion about the advantages and disadvantages of the presented approach would be interesting. The method is heuristic thus it is not guaranteed that it will give good results in all cases. 

Response 3:

We discuss the improvements and shortcomings of our algorithm in more detail in the concluding remarks of the paper.”Since the algorithm proposed in this paper only considers the static scenarios of users, in fact, users may dynamically leave the system during computing offloading or resource allocation, which affects the performance of the solution, and requires all end users to upload information to the cloud center, which will cause user information safe question. In the following research, we will study end-user mobility and combine federated learning technology with edge computing technology to protect user privacy information during task processing.”

Reviewer 3 Report

Abstract has 1 repeated sentence: "We consider the heterogeneity of mobile devices, inter-channel interference, and establish the task offloading decision of multiple end-users as a game theory-based task migration model with the objective of maximizing system utility, and propose an improved game theory-based particle swarm optimization algorithm to obtain task offloading strategies."

Authors should also consider mobility scenarios, in which offloaded task need to finish before the device moves to another location.

Authors should also estimate the impact in time/energy of running the proposed algorithms and their overall impact on the system.

When analysing results there is too much focus on utility and not so much in efficiency (this should be further discussed)

Authors should also mention standardisation activities addressing this paper topic such as ETSI-MEC

Author Response

Response to Reviewer 3 Comments 

We are very grateful to the reviewers for their comments on the article, and we have made careful revisions to the corresponding comments.

Point 1:

Abstract has 1 repeated sentence: "We consider the heterogeneity of mobile devices, inter-channel interference, and establish the task offloading decision of multiple end-users as a game theory-based task migration model with the objective of maximizing system utility, and propose an improved game theory-based particle swarm optimization algorithm to obtain task offloading strategies."

Response 1:

    We have removed duplicate content in the summary section.  

Point 2:

Authors should also consider mobility scenarios, in which offloaded task need to finish before the device moves to another location.

Response 2:

We introduce our system in static scenes and simple dynamic scenes in lines 21-25 of Section 3.1. “To facilitate experiments and obtain useful data, we consider quasi-static scenes or simple moving scenes [28-29]. In mobile scenarios, the unloading task needs to be completed before the device is moved to another location. We set the computing power of terminal devices and edge servers to be stable during task processing and may change in different time periods.” Our research group will consider complex dynamic mobile scenarios in the follow-up edge server distributed experiments.

Point 3:

Authors should also estimate the impact in time/energy of running the proposed algorithms and their overall impact on the system.

Response 3:

In terms of experimental comparison, we increase the overall system improvement rate from the comprehensive cost of the proposed algorithm's delay and energy consumption.”In addition, with the increase of computational load, the computation required for the task increases, which causes greater latency and energy consumption, and the systematic utility value decreases overall. There is a performance improvement of 15%, 10%, and 7.1% compared to TOPSO, JGORA, and ECBL schemes, respectively.” 

Point 4:

When analysing results there is too much focus on utility and not so much in efficiency (this should be further discussed)

Response 4:

    We only performed a partial efficiency comparison in the article, and we will pay more attention to the efficiency of the proposed algorithm in the follow-up experiments.

Point 5:

Authors should also mention standardization activities addressing this paper topic such as ETSI-MEC

Response 5:

We give the normalization activities addressing for MEC, the subject of this paper, in lines 14~20 of the introduction.”Mobile edge computing (MEC) is the main branch of edge computing. At present, there is no uniform definition of mobile edge computing in academia. The concept given by the European Telecommunications Standardization Institute (ETSI) is that by deploying edge servers close to mobile users at the network edge, Use the wireless access network to provide the required services and computing functions nearby. The MEC paradigm provides low-latency, mobility, and location-aware support for latency-sensitive applications.”
